# Voices of Community Partners: Perspectives Gained from Conversations of Community-Based Participatory Research Experiences

**DOI:** 10.3390/ijerph17145245

**Published:** 2020-07-21

**Authors:** Heather J. Williamson, Carmenlita Chief, Dulce Jiménez, Andria Begay, Trudie F. Milner, Shevaun Sullivan, Emma Torres, Mark Remiker, Alexandra Elvira Samarron Longorio, Samantha Sabo, Nicolette I. Teufel-Shone

**Affiliations:** 1Center for Health Equity Research, Northern Arizona University, Flagstaff, AZ 86011, USA; Carmenlita.Chief@nau.edu (C.C.); djj66@nau.edu (D.J.); abb232@nau.edu (A.B.); Mark.Remiker@nau.edu (M.R.); Alexandra.Longorio@nau.edu (A.E.S.L.); Samantha.Sabo@nau.edu (S.S.); Nicky.Teufel@nau.edu (N.I.T.-S.); 2Department of Occupational Therapy, Northern Arizona University, Phoenix, AZ 85004, USA; 3Yuma Regional Medical Center, Yuma, AZ 85364, USA; TMilner@yumaregional.org; 4Opportunity, Community & Justice for Kids, Phoenix, AZ 85027, USA; Shevaun@ocjkids.org; 5Campesinos Sin Fronteras, Somerton, AZ 85350, USA; etorres@campesinossinfronteras.org

**Keywords:** community-based participatory research, action research, community-engaged research, participatory research, community-based research, minority health

## Abstract

Community-based participatory research (CBPR) has been documented as an effective approach to research with underserved communities, particularly with racial and ethnic minority groups. However, much of the literature promoting the use of CBPR with underserved communities is written from the perspective of the researchers and not from the perspective of the community partner. The purpose of this article is to capture lessons learned from the community partners’ insight gained through their experiences with CBPR. A multi-investigator consensus method was used to qualitatively code the transcripts of a CBPR story-telling video series. Seven major themes were identified: (1) expectations for engaging in research, (2) cultural humility, (3) respecting the partnership, (4) open communication, (5) genuine commitment, (6) valuing strengths and recognizing capacities, and (7) collaborating to yield meaningful results. The themes drawn from the community partner’s voice align with the tenets of CBPR advanced in the academic literature. More opportunities to include the community voice when promoting CBPR should be undertaken to help introduce the concepts to potential community partners who may be research cautious.

## 1. Introduction

Community-based participatory research (CBPR) bridges science and practice and has been identified as an approach recommended when working with underserved and research cautious communities [1]. CBPR engages research-focused and service oriented entities in a collaborative process of co-learning, inquiry and leadership. The process can be designed to address the etiology, persistence, policies and/or behaviors perpetuating health inequities. CBPR undermines the notion of the objectivity of science and supports the goal of health equity through social justice [1,2]. Key principles of CBPR include building on the strengths of the community, establishing equitable decision making, promoting reciprocal respect and value for skills and knowledge, yielding outcomes beneficial to all partners, and creating opportunities for actions to address health issues important to the community [3]. 

CBPR has proven effective with communities striving to develop, implement and sustain culturally appropriate and relevant interventions and policies that address the social determinants of health [4,5]. A systematic literature review of CBPR’s use within clinical trials including racial and ethnic minority groups found that CBPR resulted in gains in study recruitment and retention of participants [6]. Additionally, by participating in CBPR efforts, members of diverse racial or ethnic minority groups may find satisfaction as the research outcomes could contribute to building local capacity to support health and wellness in their communities [7]. Despite the noted benefits of CBPR, the value of this approach is largely heralded through academic channels and less often conveys the perspectives of community-partners [5].

Lessons learned from CBPR provided by university-based researchers largely apply to an academic audience [8,9,10]. The literature highlights the importance of connecting the research to the larger goals of the community and clarifying roles and responsibilities for both academic and community partners [10,11]. Challenges to CBPR for academic partners include the time necessary to establish trusted relationships with the community, and to acquire required university and community institutional approvals [8,10]. In addition, CBPR projects often involve additional time to collaborate on dissemination activities. [8,9]. These perspectives do not provide the insights of community-based partners. 

The Southwest Health Equity Research Collaborative (SHERC) is a National Institute on Minority Health and Health Disparities funded (NIMHD, U54MD012388) cooperative agreement initiative of the Center for Health Equity Research at Northern Arizona University (NAU). The leadership team of SHERC includes researchers who have experience in CBPR and who work to promote the use of CBPR through SHERC’s Community Engagement Core (CEC). SHERC’s CEC facilitates university-community dialogue and action to develop collaborative research activities to promote health equity in the region. 

The CEC designs and implements educational opportunities to familiarize university scholars and community practitioners with the intent, process and outcomes of CBPR. In developing educational strategies, the CEC noted that the voice and perspectives of community practitioners explaining their reasons for working with external partners and embarking on research, a process outside the scope of their agencies, is not readily available in the literature or in training materials. To address this gap, the CEC coordinated the development of a video series of community and academic partners sharing stories of their CBPR projects. All community partners worked with underserved populations. The purpose of this review is to document lessons learned and perspectives from community partners working in underserved communities engaged in CBPR. 

## 2. Materials and Methods 

### 2.1. Participants

Members of the CEC team have experience in CBPR and were aware of local university researchers who were successfully engaged in CBPR projects. No one from the CEC team was involved as a partner in the identified research projects. In total, six CBPR projects were identified and each partnership received an email inviting them to share their stories for the SHERC CEC video series. Four pairs of academic and community partners agreed to be video recorded as a featured partnership discussing the course of their collaboration. All projects highlighted in this series were ongoing at the time of video recording and were focusing on health concerns in underserved populations impacted by adverse conditions in their social and/or physical environment. 

#### 2.1.1. Water Contamination on a Native Nation

This project is documenting the biological, chemical and psychosocial impact of a water contamination event which impacted a Native Nation in the southwest United States. The community-based partner is a municipal-level leader for a region of the Native Nation. The university partner is a water hydrologist researcher and also a member of the same Native Nation impacted by the water contamination event. 

#### 2.1.2. Environmental Contaminants and Farmworkers

This project is evaluating the health impacts of pesticides on farmworkers in southwest Arizona. Exposure to hazardous pesticides used in agricultural areas is associated with increased cancer risk, endocrine disruption, and/or reproductive-developmental toxicity [12,13]. The community-based partner is the Executive Director of a local farm worker advocacy agency who helped establish the promotora model (community lay health workers) in the region. The university partner is a biologist with expertise in environmental exposures and health.

#### 2.1.3. Bacterial Infection and Diverse Populations

This project is exploring the spread of *Staphylococcus aureus* with a regional medical center serving Hispanic and non-Hispanic residents in southwest Arizona. *Staphylococcus aureus* is a common cause of skin, soft-tissue, bone, joint, respiratory, and endovascular infections and is spread through contact with an infected host or contaminated surface [14]. The community-based partner is the Vice-President of Operations at the regional medical center. The university partner is a biologist whose research focuses on microbial pathogens.

#### 2.1.4. Young Adults Aging out of Foster Care

This project focuses on improving transition care services for youth in the foster care system as they turn eighteen and transition to independent living. The community-partner is the project manager for a non-profit agency focused on providing education and resources to youth in the foster care system in Arizona. The university partner is an occupational therapist whose research addresses functional outcomes for youth with disabilities or other co-occurring mental health disorders.

### 2.2. Data Collection

Methods used to document partners’ experiences were storytelling and guided conversation directed through a series of questions posed simultaneously to each partner. Story-telling was used as an effective means to share information with broad audiences and aligned with the CEC’s intent to use a medium accessible to non-academic audiences [15]. The questions covered five topic areas: (1) meeting and initiating the relationship including initial assumptions and expectations, (2) learning about your partner and building trust, (3) negotiating roles and responsibilities, (4) navigating challenges and successes and (5) reflections of lessons learned.

Working with Northern Arizona University’s media production services, a CEC team member scheduled partners for an in-person video recording session at the university’s recording studios. The community and academic partners received the questions prior to the session. If partners requested, a CEC team member scheduled a conference call with them up to a week before recording to prepare them for the session and answer any questions. When the partners arrived at the university studio, they were familiarized with the lighting and sound system, and sat in chairs facing each other. The CEC team member used the same question discussion guide for each video session. During the recording session, a CEC team member, off camera, posed the pre reviewed questions and partners would respond in a conversation style. The raw footage video of each partnership varied in length from 50 to 75 min. For partners traveling from out of town, SHERC covered the cost of lodging, mileage and per diem. This project was intended to produce publicly available videos, which is not research. However, the analysis of the data was considered as research and therefore was reviewed and approved by the university’s Institutional Review Board (no. 1625693-1). Each of the CBPR projects in the video series had received all required university and community-based Institutional Review Board approvals for their research projects.

### 2.3. Data Analysis

The multi-investigator consensus method was used, which applies Patton’s recommendations for non-computer-assisted qualitative data analysis [16,17,18]. The raw footage was reviewed by two analysts (master’s in public health graduate students), who were not involved in the video recording process. The analysis focused exclusively on the community partner’s perspectives. An a priori data analysis table was developed in Microsoft Excel using the five aforementioned topic areas of the discussion questions. Then the two analysts, who had not viewed the videos previously, reviewed the raw footage independently and deductively identified relevant content under each topic area, summarizing the data into themes based on the community partner’s voice only. The two analysts individually extracted quotes verbatim used by the community partners. Then they met and read through the content extracted and came to a group consensus upon identified patterns, defined as recurring concepts or ideas that revealed a descriptive trend. The two analysts then reviewed their preliminary findings with another member of CEC team not previously involved in the analysis. As a group they collectively reviewed the patterns, discussed these and came to consensus on the final identified themes. In addition, the analysts selected 2–4 quotes per theme to support interpretation of content analysis and created summary results. These summary results were member checked with community partner interviewees via their review and contribution to this article.

## 3. Results

After conducting consensus analysis of the four videos, seven major themes that cross cut the predetermined topic areas were identified: (1) expectations for engaging in research, (2) cultural humility, (3) respecting the partnership, (4) open communication, (5) genuine commitment, (6) valuing strengths and recognizing capacities, and (7) collaborating to yield meaningful results. The themes align with the tenants of CBPR and highlight that CBPR principles touted in the academic literature are valued by community partners engaged in collaborative research. 

### 3.1. Expectations for Engaging in Research

Expectations for engaging in research varied among community partners, revealing hesitation, apprehension, hope, and/or excitement at the prospect of engaging in research with an academic partner. Some partners conveyed feelings of apprehension regarding research, often due to poor experiences with previous research or a general distrust. For example, from the water contamination partnership, the community partner shared concerns about research rooted in a historical perspective that their Native Nation community had been over studied and exploited.

“Apprehension goes back to our history where we have been studied and studied and researched many times without our consent … many people have come into our communities to do all manner of research on our lives, on our health, lifestyles, on our culture, etc., and we don’t see them again, they go off back to wherever they came from and I imagine profit off whatever they did out here and we don’t hear back from them.”

The community partner focused on farmworker justice from the environmental contaminants also had dissenting feelings regarding research. Her hesitation when approached by academic researchers related to what the unknown motivations were of the researcher and whether or not they aligned with what was needed in her community. She also had to decide if she had the time to take on the project. In other partnerships, such as the bacterial infection and young adults aging out of foster care, the community partners expressed more positive outlooks regarding research. The Vice-President of Operations at the medical center from the bacterial infection partnership had previous good experiences with researchers and the university and shared a favorable perspective. The Project Manager at the foster care agency had not had experience with research and was hopeful and excited about the opportunity to engage in research because she hoped that research could have a beneficial impact on the foster care youth the agency serves.

“When we first started working on the research project, I didn’t know what to expect, I was going along for the ride knowing that I’m going to get something for these kids whatever that would be ... so when you came along and said oh let’s do this, I’m like I think I took a sigh of relief, yes finally more resources for these kids because they’re alone.”

### 3.2. Cultural Humility

Understanding and appreciating different cultures, or cultural humility, was identified as one of the key principles to building trust among partners and having a successful partnership. Cultural humility involves practicing ongoing self-evaluation and self-critique regarding one’s own cultural understandings and how those perspectives can impact the research process [19]. Community partners shared the importance of recognizing and addressing language barriers as one of the ways to be culturally humble in a partnership. Language barriers can include partners’ primary language, but also terminology and jargon used in a research context and local expressions used in communities. For the community partners, simplifying or adapting the research language to ensure their community’s understanding was imperative to fully connect to the project and effectively disseminate information to their respective communities. The Native Nation municipal-level leader described how the academic partner’s expression of cultural humility gave the community reassurance in their partnership:

“You [academic partner] were able to present all of that in our own language to our community in such a way that they understood it, that they could conceptualize even your scientific research. You put it into terms that grandma sheepherder, grandma farmworker could understand. So I think that once I saw that in you as a person ... I was reassured.”

Cultural humility was also represented in taking clear and evident steps to demonstrate to the community partner that there was respect for the community’s knowledge. The Vice-President of Operations at the medical center recognized that the university partner was willing to hear from stakeholders in the community directly, which demonstrated humility and a commitment to establishing a solid collaboration. 

“Historically there have been partnerships that don’t work well because people don’t set the right foundation ... in our case, what I appreciated most was that you came and talked to us directly about your work, but also about the respect that you and other researchers had for the people of our community ... your willingness to talk directly and engage the key stakeholders ... this created a strong collaboration.”

### 3.3. Respecting the Partnership

Community partners reflected on how respecting the partnership contributed to their partnership’s success, noting that valuing and listening to one another was essential for academic-community collaboration. This quality required making an effort to get to know each other and understand each other’s work. Having an understanding of both partners’ perspectives fostered trust in the research and the partnership. In the foster care partnership, the community partner felt that having this fundamental respect and understanding resulted in a stronger partnership that could accomplish more: 

“As an organization, as a nonprofit, working with someone who’s in research. doing the research portion, it’s important for us to have an understanding of where you’re coming from and having a timeline in our head so we have dates so we’re not trying to push it too fast is always important to know and to keep in mind as well so that our successes can be bigger and not mediocre. The more we work together, the more we listen to each other and the more we understand both sides, the stronger we’ll be.”

The Executive Director of the farmworker advocacy agency explained about respecting the partnership as being open with one another, having respect for one another, and valuing each member of the team as equal partners.

“It’s about really being open with each other ... listening to each other and respecting each other as equal partners and valuing the information that each brings.”

### 3.4. Open Communication

More than simply listening to each other, open communication constituted both partners being intentional about communicating and being flexible when finding solutions to challenges that arose throughout the partnership. Community partners valued transparency in the research project and the partnership, alluding to the importance of honesty and patience in communicating when working through issues together. Their effective communication strategies created visible rapport with one another as evidenced by their sharing laughter and confirmatory gestures (i.e. head nodding), with one another throughout the discussion. The Executive Director of the local farmworker advocacy agency favorably recalled her experience working with her academic partner through a challenge encountered in the research process.

“I think I’m always very honest and open, and I made it known to you guys. But you guys were always very patient, very nice, and very like calming, and ‘okay thank you for what you’re doing,’ or ‘we appreciate what you’re doing.’ You heard me.”

Open communication was also evidence by the university partner offering different formats for communication in order to reach all stakeholders. For some community members, an email may be enough, but for others there was a need to host a direct conversation. This sentiment was expressed by the Vice-President of Operations at the medical center.

“You [academic partner] did a great job not only communicating with me, but communicating with our entire team—whether it was by emails, or you took the time to have a conference call with all team members so that everybody could hear the information at the same time. I think that that has significantly contributed to the quality of the relationship that we’ve developed.”

### 3.5. Genuine Commitment

Genuine commitment, defined as a true and honest desire to help, facilitates trust-building and is one of the themes that was relevant among the four partnerships. Community partners were able to recognize their academic partners’ sincere commitment to the people and the community’s well-being, which fostered trust among the partners. Community partners also appreciated the enthusiasm and dedication of their academic partners, communicating that the partnerships were more prosperous when researchers shared responsibility and an aspiration to fulfill project goals. The Vice-President of Operations at the medical center shared the following sentiment regarding genuine commitment.

“This wasn’t just a researcher reaching out trying to find a site, a place where he or she might conduct research, but there has always been a true commitment to the people of the community. In a community-based research project, that is one of the most fundamental and important elements of making a successful partnership.”

Additionally, the Native Nation municipal-level leader identified that the university researcher showed a sincere interest in and caring for the community. The university partner’s demonstrated a genuine level of commitment through her honesty.

“I saw that there was sincerity with you. Yes very much, the compassion that you have innately for our people and for our land, for our water … we need sincerity, we need honesty, we need objectivity, and we need truth you know.”

### 3.6. Valuing Strengths and Recognizing Capacities

From the community partner’s perspective, valuing strengths and aligning the needs of the project with each partner’s capacities was integral to building a successful partnership. Exchanging information on each partner’s strengths and capacity levels allowed the partners to adopt roles and responsibilities based on their capabilities and needs to move the research project forward. The Vice-President of Operations at the medical center describes this theme after reminiscing about how within their partnership they addressed capacities and made adjustments as they saw fit without jeopardizing their workload or creating additional burdens. The open relationship that was created allowed the partnership to be cognizant of partner strengths and to move forward with their project effectively and efficiently.

“Really aligning what your needs were, what our capacity was, and talking about processes that would make us both successful, was a very important part of the process of building an effective partnership.”

Community partners shared the importance of recognizing, understanding, and communicating the limitations and capacities within the partnership to engage in research. The Project Manager at the foster care agency acknowledged their limited capacity prior to this project for engaging in research and also the importance of being honest about recognizing each partner’s skills and abilities.

“I think it was just a great back and forth of ‘oh this needs to be done’, ‘oh I can do that’ ‘oh okay, so this needs to be done’ ‘okay what else do you need?’ and it’s just finding out who knew what to do what. You guys guided me very well because this is my first time doing a research project, ever. So going through those classes and learning consent, and yes it was dry reading, I totally get it, but it was very knowledgeable and it was very important for me to know your boundaries and the processes you guys had to go through on your side ... I think that was key.”

### 3.7. Collaborating to Yield Meaningful Results

Community partners from the projects discussed their journey in their partnership with the academic researcher from the first encounter to reaching some of their goals. Collaborating to yield meaningful results was the final theme identified, capturing the notion that working together on a CBPR project means intentionality in producing beneficial outcomes such as community empowerment, visibility, and reassurance. From the water contamination partnership, the Native Nation municipal-level leader expressed gratitude for the results of the research project and explained how the information and data gathered brought reassurance back to the community regarding a fear of or concern about water contamination.

“The actual great positive that came out of it was the study research results that you and your team generated which showed, demonstrated very clearly that the contamination levels of the river should not be a concern, they are very minimal and the emphasis that was made there and you articulating those results is working in a very grand way by the people and the farmers—that apprehension of using the water is being lifted and we’re beginning to farm again and the work that you and your team did greatly contributed to addressing that stigma that developed ... your work gave us the reassurance that we’re okay.”

The Executive Director at the farmworker advocacy agency communicated that the research results had an influential and empowering impact on the community because it simplified scientific data so it could be easily understood and disseminated within her community. The results provided reassurance to the community partner that farm workers would become more aware of environmental contaminants and take precautions to ensure their safety. The Vice-President of Operations in the medical center engaged in the bacterial infection partnership offered sentiments of appreciation for research collaborations with an academic partner. She recognized the value of community-based partnerships, especially in allowing the community to voice their thoughts and concerns and to play a responsible and equitable role. She identified that the project could help the medical center identify ways they could change how they deliver care to avoid the spread of *Staphylococcus aureus.*

Similarly, the Project Manager at the foster care agency expressed gratitude and enthusiasm for research partnerships that could help her agency better serve the community. This community partner found value in the collaboration, sharing that this particular partnership helped foster collaboration with other organizations who serve the same community and allowed her organization to gain visibility with state officials.

“Because of the research we’ve already done, [our agency] and other organizations working with the same population are starting to open up a little bit more and are willing to collaborate more. We’re willing to sit around the table, find out what everybody is doing so we can work together for these kids ... I’m very thankful for the day you introduced research to us because it didn’t just help us, it helped the population that we served. It helps us as an agency with grants because you’re helping us get those deliverables, those measurables … the research part comes in and gives us tangibles that why wouldn’t we work with you?”

## 4. Discussion

This review offers insights from four community partners engaged in CBPR. Too often, strategies for best practices in CBPR are from the voices of university-based researchers and not the community partner [8,9,10,11]. Reflections on CBPR from the university perspective highlight the importance of collaborative and locally relevant outcomes, while also discussing challenges specific to a university audience [8,9,10,11]. This project is novel in documenting the voices of four community partners’ perspectives on CBPR, a point of view that is needed to support the success of CBPR partnerships. 

Community partners discussed trust, cultural humility, valuing diverse skills sets, respect, open communication, and commitment to the research as vital to the success of a CBPR project. These values are the very foundation of CBPR as described in the academically based peer reviewed literature [1]. Similar to the research literature, the community partners also highlighted the need for partnerships to yield meaningful results to the community, as this builds trust and also the willingness to participate in future collaborations [11]. It is also critical to recognize strengths and capacities of each partner so roles and responsibilities could be appropriately assigned [3]. Demonstrating genuine commitment to the partnership was characterized as the university partner demonstrating a personal investment in the communities impacted by the CBPR partnership. Items discussed in the academic literature that were not highlighted by the community partners were challenges associated with meeting project timelines, ensuring equitable compensation for community partners, and expectations for disseminating study results in the academic literature [8,10].

To the credit of the featured collaborations, the community and academic partners were able to develop a respectful, effective process. Experienced community partners have been underutilized advocates and teachers of CBPR. There is a need to further gather community member perspectives on engaging in CBPR efforts [20]. University sponsored open access journals which promote the inclusion of community members in research development and implementation are available open access and online [21,22]. Toolkits providing step by step guidance for communities on CBPR are also available [23]. However, these journals and toolkits are led primarily by university-based teams and perhaps lack a true community perspective. The perspectives documented in these videos suggest community partners in CBPR projects should be collaborating on the development and dissemination of educational resources intended to encourage their involvement in CBPR.

The results of this analysis are non-generalizable but could guide future research to identify the concerns communities may have about engaging in research efforts. The more local community members were familiar with the pros and cons of CBPR, the more they would see their academic neighbors as a potential resource to help address severe health inequities needing innovative and evidence-based approaches. Communities cautious of research have experienced or are familiar with losing control of the process. By hearing from their peers in community health about CBPR and the opportunity to share control and decision making with academic partners, local leaders and community members may be willing to engage in alternate ways to address acute and chronic health concerns.

### Strengths and Limitations

A limitation of this study was that it only included long-standing partnerships. It is possible that, due to the considerable maturity of these partnerships, they had resolved issues related to power dynamics, which may be more evident in newer CBPR partnerships. The project could have been improved by including a reflexive-oriented question which asks both partners to speak directly to how they confronted external and internalized power dimensions within their partnerships early on [5,24]. Additionally, contextual information from dialogue shared between partners could have been missed by only analyzing the statements of the community partners. The video sessions were recorded at the university’s media studios which could have influenced respondents’ discussion. Therefore, future efforts should be made to capture this type of data within the community. However, a strength of these videoed conversations and their analysis is the potential for community partners to be advocates of the approach and provide practice-based evidence that outcomes can be mutually beneficial and contribute to capacity building in communities and academia.

## 5. Conclusions

The perspectives of these four partners highlight the importance of clear expectations, cultural humility, respect, open communication, genuine commitment, valuing the strengths and capacities of each partner, and producing meaningful results to the community when engaging in CBPR efforts. The reflections from these community partners mirrored many of the key principles of CBPR [3]. The principles of CBPR shared by community partners in this study, can be highlighted in education and advocacy efforts supporting CBPR. To learn more about the partnerships in this study please visit the Stories of Community-Engaged Research website, https://nau.edu/sherc/cec-video-series-landing-page/ [25].

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
