# Peer review of "Voices of Community Partners: Perspectives Gained from Conversations of Community-Based Participatory Research Experiences"

_ijerph, 2020, doi:10.3390/ijerph17145245_

Round 1

Reviewer 1 Report

The purpose of this manuscript was to glean insights from community partner’s experiences with community-based participatory research (CBPR).  As identified by the authors, there is a lack of research exploring community partner perspectives in CBPR projects and this article has the potential to make an important contribution in this regard. This manuscript is strengthened by the use of novel methods like storytelling, which the authors point out may facilitate communication with non-academic audiences. Furthermore, the authors involved several researchers in a consensus-based analysis process, demonstrating thoughtful attention to the rigour of their analysis.  In addition to the specific feedback provided below, there are some overarching revisions required before this manuscript can be recommended for publication. For example the introduction would benefit from a paragraph outlining the focus on academic partner perspectives in CBPR in the literature. Also, the relationship between CEC and the highlighted projects is unclear; an explanation is needed of how the specific partnerships were chosen/recruited and how community members in particular, were involved in the current study. The authors claim that IRB clearance was not required because the videos are in the public domain, but if the intent was to use the videos as data (and the ‘raw footage’ – line 131- was analysed), shouldn’t informed consent by those involved have been sought? Given the similarity between the questions topic areas (lines 80-83) the resulting themes, the authors should clarify how they avoided confirmation bias (e.g. was an inductive approach used in the process of identifying patterns)? The discussion spends too much space continuing to justify the study (rationale should be moved to introduction) and would benefit from a brief summary of the results in relation to available literature for all seven themes. The discussion would also be strengthened by acknowledging study strengths and limitations.

Introduction

Overall: Given that the purpose of the introduction is to build a rationale for the study, the authors should consider adding a paragraph after line 54 that exemplifies the academic-focused nature of CBPR perspectives. Articles like Castleden et al., (Castleden, Heather, Vanessa Sloan Morgan, and Christopher Lamb. "“I spent the first year drinking tea”: Exploring Canadian university researchers’ perspectives on community‐based participatory research involving Indigenous peoples." The Canadian Geographer/Le Géographe canadien 56.2 (2012): 160-179) can provide this rationale. In consideration of word limits, miscellaneous information about the SHERC (line 54-60) could be removed. The introduction should also clearly state the study purpose.

Page 2, Line 44: The sentence “Practiced since the 1990s in the field of public health, CBPR …” makes it seem that CBPR has only existed since the 1990’s – this sentence should be revised to make clear that participatory/action/community-engaged research is not a new approach.

Page 2, Line 46: Fix referencing by changing [4] [5] to [4,5].

Page 2, Line 58: Change “of” to “in”.

Page 2, Lines 65-66:  the sentence “the voice and perspectives of community practitioners explaining their reasons for working with external partners and embarking on research, a process outside the scope of their agencies, is not readily available in the literature or in training materials” seems to suggest a rationale for the current study, yet the focus of the research does not appear to be about exploring community reasons for working with external partners (a WHY question) -  rather, the questions focused on initiating the partnership, assumptions and expectations,  building trust,  negotiating roles and responsibilities, navigating challenges and successes - all HOW questions; the study rationale and purpose should be clarified.

Page 2, Line 69-71: This sentence is about the Methods and should be removed here.

Materials and Methods

Overall: Considering that this manuscript deals with CBPR projects, it is unclear if/how participants were given an opportunity to participate in the research process. Furthermore, it is unclear what relationship the CEC has to the projects being highlighted. More information is needed in the Participants section about how the pairs of academic and community partners were recruited. This section may be better placed at the beginning of the Materials and Methods section prior to the Data Collection section. The rigour of the qualitative analysis should also be described.

Page 2, Line 75: Videotaped should be changed to video recorded to reflect the technology used for recording unless tapes were used. These changes should be made throughout the methods section. 

Page 2, Line 74 to Page 3, Line 95:  It should be made clear that the authors are focusing on the stories from community partners.

Page 2, Line 85: Remove “90 minute” as it may be confused with the length of actual sessions later described on page 3, line 92.

Page 3, Line 100: The Water contamination on a Native Nation should be a Level 3 subheading just like all of the other projects being described. Is Water contamination on a Native Nation the community’s preferred title of the project?

Page 3, Line 100-125: Information provided about each of the projects differs slightly between projects. For example, the description of the “Water contamination on a Native Nation” includes the approach (mixed methods) that does not appear for any of the other studies. Miscellaneous information should be removed for clarity.

Page 4, Line 128: Add “a” after “A…”

Page 4, Line 126: Were community partners given an opportunity to take part in data analysis consistent with the CBPR approach of the studies they were involved in?

Page 4, Line 132-133: If academic and community partners “briefly discussed their intended responses”, how were data from one partner analyzed without reviewing the data in its entirety? Research on group-based data collection suggests that data from one interviewee often contextualises data from the other interviewee (Gibbs, 2012).

Page 4, Line 134: The multi-investigator consensus method of qualitative analysis is unclear. This method should be described in more detail. How was consensus achieved? How were disagreements dealt with?

Results

Overall: Some of the themes (e.g. expectations for engaging in research, respecting the partnership) were very similar to the questions posed to interviewees (e.g. meeting and initiating the relationship including initial assumptions and expectations, negotiating roles and responsibilities). How did researchers ensure that there wasn’t confirmation bias in the analysis of results? An inductive analysis approach may help alleviate these concerns.

Page 4, Line 146 to Page 5, Line 181: There is an over reliance on quotes in this description of the theme. The authors could focus on key example quotes rather than highlighting quotes from each project.

Page 5, Line 182-198: You stated in Methods and Materials that “analysts selected 2-4 quotes per theme to support 137 interpretation of content analysis”, yet this theme (and others) only included a single quote. It is unclear why some themes are more thoroughly explained than others. 

Page 7, Line 289: Remove “who had not previously engaged in research with an academic partner” as it is repetitive with information previously presented in Methods and Materials.

Discussion

Overall: Too much space is spent justifying the study and it is repetitive with information presented in the Introduction (e.g. “There is a need to further gather community member perspectives on engaging in CBPR efforts [16].”). Instead, the novelty of the community perspective should be discussed. More space should also be dedicated to an expanded summary of the results, with direct comparisons with available literature for each of the seven themes. The discussion should also explore the strengths and limitations of the study design, some of which appears in the Conclusions starting on Page 8, Line 331.

Page 8, Line 301-302: This sentence should be replaced with the stated purpose of the study for clarity.

Page 8, Line 304-305: Remove this sentence as it is repetitive with information previously presented in Methods and Materials.

Page 8, Line 313-318: It is unclear how the availability of resources relates to the results from the study.

Conclusions

Overall: The conclusion would be strengthened by highlighting key findings and by providing an explicit discussion of future research opportunities.

Page 8, Line 322-323: Why is this question being posed? Rather than focusing on the mediums used to communicate the benefits of CBPR to community members, the conclusion would be strengthened by highlighting research needs.

Page 9, Line 334-336 Is this recommendation based on the authors belief that having academic partners present in the interview influenced the community partner’s responses? This should be part of any discussion about strengths and limitations of the study so that it is clear to the reader how this recommendation emerged.

Page 9, Line 336 – 338: It is unclear how/why this recommendation emerged.

Page 9, Line 339: The term community researchers is introduced for the first time. Previously these persons were referred to as community partners. Revise for consistency.

Reviewer 2 Report

I really enjoyed this paper, it was original, creative and added to the literature on CBPR partnerships. I've outlined suggested changes in the attachment.

Reviewer 3 Report

The research in CBPR is an academic goal that may lead to policy, systems, and environmental change in community. This is an excellent and much needed paper on the  "voice and perspectives" of community partners.  Questions of the intent to collaborate and the journey is much needed in the literature.  My feedback below are suggestions to further enrich the paper.

The authors recognize that while academic and community partners are in partnership there are different perspectives, language and values along the journey.

The innovation of this articles comes from the methodology. Using story-telling video series is a unique way to capture the community partners voice.  The article could be improved in three ways.

First, the article could link to the video content as an additional resource so readers can learn from the experience by watching one or more of the videos.

Second, the paper could improve by describing the power dynamics embedded in the relationships between community and academic partners. This can be done in the introduction section relating the literature, in the description of the partnership, or in the discussion section related to the results. 

It is surprising that the power dynamics were not discussed by community partners. Do the authors feel power is balanced between these community and academic partners e.g. Vice President of Operations, Executive Director farm worker advocacy? Alternatively, do the authors feel the power is balanced because of the story-telling methodology?

Third, the paper could describe the emotion and non-verbal insights from the video in the results section. A few brief insights on the affect and non-verbal reactions of participants could complement the quotes and speak to comfort, confidence, joy and balanced power where appropriate. 

Much needed and innovative article to advance the direction of CBPR work.

Round 2

Reviewer 1 Report

Some aspects of the manuscript are improved - in particular, the introduction and methods sections have been strengthened. However, there are still areas of weakness and a major concern regarding the author response to the question about the requirement for IRB approval (also noted by the second reviewer). The egregious history of research involving Indigenous communities can only be remedied by exemplary adherence to the highest ethical standards by researchers who work with Indigenous communities. My main concerns are:

  • The justification for skipping an ethics review and approval process is weak. While it is understood that the data were not initially collected for research purposes, the video footage was treated as data. The Northern Arizona University IRB stipulates that 1) researchers do not have the authority to determine if they need IRB approval or exemption, and 2) data collected initially for non-research purposes that are later used for research purposes require IRB approval when identifiable human participants are involved.  Further, the use of data (especially within a CBPR context) should typically be vetted by the community of interest. Many communities have their own ethics’ review process; the authors should clarify if any of the projects discussed in this manuscript were working with local community-based research ethics’ boards and if these boards were consulted about the current research.
  • The description of analysis implies that a deductive – not inductive – approach was used.
  • The themes are not well-supported by the quotes included in the manuscript.
  • The discussion focuses on summarizing rather than comparing the findings with existing literature and the recommendations/conclusions are not supported by the data collected. 
  • The writing is weak in some areas. There are discussion points in the Conclusion and conclusions in the Discussion.
  • The overall message in the conclusion is disappointing given the rich content within the results section and the novelty of the perspective being shared.

More specifically:

Page 2, Line 59: Change “produce collaborative results reports…” to collaborate on dissemination activities…” for clarity.

Page 2, Line 75: Change “perspective” to “perspectives”.

Page 4, Line 139: Change “taping” to “recording” to reflect the technology used in data collection.

Page 4, Line 148-158: Given that the analysts used an a priori data analysis table, this reviewer is still unclear on how this represents an inductive analysis approach (as indicated on Line 151) given that the table used by the two data analysts was developed using the topic areas identified during data collection. The word “inductively” should be changed to “deductively”.

Page 4, Line 158: Given that authors identified 2-4 themes to “support interpretation”, it is still unclear why only one quote is used to support 5 of the 7 themes. This should be changed to “one to four quotes” or these additional quotes should be added to support the interpretation of the data as stated in the initial review.

Page 8, Line 311-319: As stated in the initial review, this paragraph should highlight the novelty of focusing on the community perspective.

Page 8, Line 320-326: This paragraph would benefit from identifying if the community partner perspectives are similar to, or differ from, university-based partner perspectives in the literature.

Page 9, Line 357-359: It is still unclear how/why this recommendation emerged based on the identified themes.

Author Response

Please see the PDF attachment response to reviewer table.

Round 3

Reviewer 1 Report

The manuscript is much improved. Thank you for your detailed attention to reviewer comments. I especially appreciated your checking in with your university's REB. Given the focus of the topic and the readership you are likely to attract, the example you are providing is commendable.